Sequence characteristics, expression and subcellular localization of PtCYP721A57 gene from cytochrome P450 family in Polygala tenuifolia willd

Luo Yao
Hu Benxiang
Ji Haiyue
Jing Yiyao
Zhang Gang
Yan Yonggang
Yang Bingyue bingyyang@126.com
Peng Liang ppengliang@126.com
College of Pharmacy, Shaanxi University of Chinese Medicine/Shaanxi Engineering Research Center for Application and Development of Chinese Medicine in Qinling Mountains/Key Laboratory of “Qin Medicine” Research and Development , Xianyang , China
Uversky Vladimir
Electronic publication date: 2024 Oct 18
Publication date: 2024
Volume: 12
Electronic Location ID: e18089
Received 2024 Jun 3; Accepted 2024 Aug 22
Copyright: © 2024 Luo et al.
Copyright year: 2024
Copyright holder: Luo et al.
License: This is an open access article distributed under the terms of the Creative Commons Attribution License, which permits unrestricted use, distribution, reproduction and adaptation in any medium and for any purpose provided that it is properly attributed. For attribution, the original author(s), title, publication source (PeerJ) and either DOI or URL of the article must be cited.
License URL: https://creativecommons.org/licenses/by/4.0/

Keywords: Polygala tenuifolia, Cytochrome P450, Gene cloning, Subcellular localization, Gene expression

Funding: National Natural Science Foundation of China 82003899 General Project of Social Development of Shaanxi Provincial Department of Science and Technology 2024SF-YBXM-457 Youth Project of General Project of Shaanxi Provincial Science and Technology Department 2024JC-YBQN-0989 Yulin Science and Technology Plan Project YF-2021-74, YLKG-2023-17 Subject Innovation Team of Quality Control and Resources Development of “Qin drug” of Shaanxi University of Chinese Medicine 2019-QN01 This work was supported by the National Natural Science Foundation of China (Grant No. 82003899), the General Project of Social Development of Shaanxi Provincial Department of Science and Technology (2024SF-YBXM-457), the Youth Project of General Project of Shaanxi Provincial Science and Technology Department (2024JC-YBQN-0989), the Yulin Science and Technology Plan Project (YF-2021-74, YLKG-2023-17), and the Subject Innovation Team of Quality Control and Resources Development of “Qin drug” of Shaanxi University of Chinese Medicine (2019-QN01). The funders had no role in study design, data collection and analysis, decision to publish, or preparation of the manuscript.

==============================
The Cytochrome P450 (CYP450) family is the largest enzyme protein family in plants, distributed across various organs and involved in significant catalytic activities in primary and secondary metabolic processes. In this study, we cloned the PtCYP721A57 gene, characterized its open reading frame (ORF), and conducted comprehensive analyses including physicochemical properties, evolutionary relationships, subcellular localization, prokaryotic expression, and correlation between the relative expression of different parts and the content of tenuifolin, hormones, and abiotic stress response associated with the encoded protein. The ORF of PtCYP721A57 was 1,521 bp, with a secondary structure predominantly composed of α-helices and random coils. Subcellular localization experiments confirmed the presence of PtCYP721A57 in the endoplasmic reticulum. For prokaryotic expression, we constructed the recombinant plasmid pET28a-PtCYP721A57 using pET28a as the vector, which was then transformed into BL21(DE3). Induction with Isopropyl β-D-1-thiogalactopyranoside (IPTG) at temperatures of 16 and 25 °C and varying concentrations (0.1, 0.2, 0.5, 1, 2 mM) resulted in the formation of inclusion bodies, with higher expression observed at 25 °C. Our qPCR analyses revealed that PtCYP721A57 exhibited the highest expression in the cortex of Polygala tenuifolia, followed by roots and xylem, correlating with the observed tenuifolin content distribution. Induction with abscisic acid (ABA) and chitosan (CHT) initially decreased PtCYP721A57 expression followed by a subsequent increase, peaking at 48 h. Similarly, drought stress induced a gradual increase in PtCYP721A57 expression, also peaking at 48 h. NaCl treatment for 6 h significantly upregulated PtCYP721A57 expression. In conclusion, our study provides foundational insights into the PtCYP721A57 gene in Polygala tenuifolia, laying the groundwork for further exploration of its role in the biosynthesis pathway of triterpenoid saponins.

Introduction

Polygala Radix, a traditional Chinese medicine derived from Polygala tenuifolia Willd., is characterized by its “bitter, hot, and warm properties”, and “belongs to the heart, kidney, and lung meridians”. Originally documented in the “Shennong Herbal Classics”, it is renowned in the Chinese Pharmacopeia for its “tranquilizing, enlightening, phlegm-dispelling, and swelling-reducing effects.” Clinically, it is used for palpitations, insomnia, amnesia, epilepsy, cough with phlegm, carbuncles, sore toxins, and breast swelling and pain (Sun et al., 2023). This medicinal herb is among the 85 traditional bulk medicinal materials for export and is one of 42 tertiary wild varieties under key protection in China (Xue et al., 2015). It ranks among the top three single-flavor Chinese herbs in clinical nootropic prescriptions and is considered essential for health maintenance. Studies have identified over 140 compounds from P. tenuifolia, primarily including saponins, ketones, oligosaccharide esters, alkaloids, and other bioactive components. These compounds and extracts exhibit a broad spectrum of pharmacological activities, such as anti-aging, neuroprotective, antidepressant, sedative-hypnotic, anti-inflammatory, antiviral, anticancer, antioxidant, and antiarrhythmic effects (Yao et al., 2022). P. tenuifolia is widely distributed in the Northwest, North, and Northeast regions of China, with particularly abundant and high-quality sources found in Shaanxi and Shanxi Provinces. It is classified as a moderate xerophyte with strong drought resistance but susceptibility to waterlogging (Peng et al., 2018).

The primary active ingredient in P. tenuifolia is oleanane-type triterpenoid saponins, which exhibit pharmacological actions such as enhancing learning and memory, sedation, and antidepressant effects (Lacaille-Dubois, Delaude & Mitaine-Offer, 2020). In higher plants, the biosynthesis of oleanane triterpenoids occurs in three stages: precursor supply, skeleton synthesis, and terpenoid synthesis. During the third stage, β-amyrin synthase (β-AS) catalyzes the conversion of 2,3-oxidosqualene (OS) to β-amyrin. Under the influence of Cytochrome P450 (CYP450) enzymes, β-amyrin undergoes oxidation at C-28 to form oleanolic acid. Subsequently, oleanolic acid is further modified by CYP450s and UDP-glucuronosyltransferases to produce a diverse array of triterpenoids (Sandeep et al., 2019; Fanani et al., 2019).

CYP450 is the largest family of enzymes in plants (Liu et al., 2022), characterized by a cysteine-iron heme structure that enables the reduction of oxygen molecules and a wide array of catalytic activities. It plays a crucial role in the biosynthesis of terpenoids, alkaloids, and phytohormones (Yu et al., 2017). Structural modifications catalyzed by CYP450 are pivotal for the diversification and functionalization of triterpenoid saponins, making it a key enzyme in the biosynthetic pathway of these components (Mak & Denisov, 2018). P450 genes also contribute to plant growth and development, enhancing tolerance to both biotic and abiotic stressors (Laffaru & Sharma, 2021).

Yang et al. (2018) demonstrated that the CYP450 enzyme in Betula platyphylla exhibits tissue-specific and hormone-induced expression characteristics, potentially playing a crucial role in the growth, development, stress resistance, and metabolite synthesis of Betula platyphylla. The identification and functional verification of key CYP450 genes involved in the biosynthetic pathway of triterpenoid saponins in P. tenuifolia, along with the application of plant metabolic engineering to enhance triterpenoid saponin content, have significant theoretical and practical implications. This study utilized bioinformatics analysis of PtCYP721A57, combining protein subcellular localization, prokaryotic expression, response to hormones and abiotic stress, relative expression across different plant parts, and accumulation of tenuifolin content to elucidate its role in the biosynthesis pathway of tenuifolin. It may provide insights into the crucial role of the CYP450 gene in the defense responses of P. tenuifolia, offer evidence for understanding the biosynthesis pathway and regulatory mechanisms of tenuifolin components, and establish a foundation for enhancing the quality of medicinal materials through genetic engineering. This research laid a basis for further investigations into the functional aspects of the CYP450 gene in P. tenuifolia.

Materials and Methods

Experimental materials

P. tenuifolia seeds and plants were collected from the Pharmaceutical Botanical Garden of Shaanxi University of Traditional Chinese Medicine (Xianyang, Shaanxi Province) (108°74′E, 34°31′N). Three strains of 3-year-old P. tenuifolia with consistent growth were selected, and the roots, stems, and leaves were pooled for full-length transcriptome sequencing analysis. Additionally, different parts (roots, xylem, cortex) were collected under uniform conditions for further study, in conjunction with previous experimental results from our research group and related studies. Thirty-day-old seedlings of P. tenuifolia, showing robust growth and uniform developmental status, were selected for the following treatments:

Seedlings were exposed to 200 μmol·L−1 chitosan (CHT), 200 μmol·L−1 abscisic acid (ABA), 100 mmol·L−1 NaCl, and 10% PEG6000 for durations of 0, 6, 12, 24 and 48 h. Untreated seedlings served as the blank control.

Amplification of the PtCYP721A57

The total RNA from P. tenuifolia seedlings was extracted using the Trizol extraction kit from Bioengineering Co., Ltd. (Shanghai, China), following the manufacturer’s protocol. The extracted RNA was then reverse transcribed into cDNA using the PrimeScript™ II 1st strand cDNA Synthesis Kit with Reverse Transcriptase from Takara Co., Ltd. (Japan). Specific primers designed for the PtCYP721A57 gene sequence (CYP721A57-1-F/R, as listed in Table 1) were used for PCR amplification with the cDNA as a template. The resulting PCR products were analyzed by 1.0% agarose gel electrophoresis. The target DNA fragment was purified and recovered using a DNA gel recovery kit, ligated with pMD™19-T vector and transformed into DH5α competent cells. Positive clones were selected and sent to Aoke Biotechnology Co., Ltd. (Yangling, China) for sequencing.

Table 1 Primer sequences for gene cloning and expression analyses of PtCYP721A57.

Primer	Sequence (5′→3′)	Usage	Length/bp	
PtCYP721A57-1	F: ATGATGCAGTTCCTCTTAGCT	ORF amplification	1,521	
R: TCAATCACGAAGTTTCGTGA	
PtCYP721A57-2	F: CGGGATCCATGATGCAGTTCCTCTTAGCTG	ORF amplification of prokaryotic expression	1,521	
R: CCCAAGCTTTCAATCACGAAGTTTCGTGAAGATG	
q-PtCYP721A57	F: AAGGGATCAAAGGACCTGGC	q-PCR	152	
R: TGGTGCCCACTTGTGGTAAA	
GADPH	F: ACAGCAACGTGCTTCTCACC	q-PCR	128	
R: CCCTTCATCACCACCGACTA	

Bioinformatics analysis of the PtCYP721A57 protein

Bioinformatics analysis and prediction of PtCYP721A57 were conducted using online bioinformatics software tools (Table 2).

Table 2 Bioinformatics software and websites for analyses of PtCYP721A57.

Biological information	Related software and website	
Open reading frame	https://www.ncbi.nlm.nih.gov/	
Prediction of physical and chemical properties	ProtParam (https://web.expasy.org/protparam/)	
Phosphorylation site prediction	NetPhos 3.1 Serve (https://services.healthtech.dtu.dk/services/NetPhos-3.1/)	
Protein structural domain prediction	ExPASy (https://prosite.expasy.org/prosite.html)	
Prediction of secondary structure	SOPMA (https://npsa-prabi.ibcp.fr/cgi-bin/npsa_automat.pl?page=npsa%20_sopma.html)	
Prediction of tertiary structure	SWISS-MODEL (https://swissmodel.expasy.org/interactive/)	
Protein signal peptide prediction	SignalP-5.0 (https://services.healthtech.dtu.dk/service.php?SignalP-5.0)	
Transmembrane information	TMHMM (https://services.healthtech.dtu.dk/service.php?TMHMM-2.0)	
Phylogenetic tree construction and beautification	MEGA11 Chiplot (https://www.chiplot.online/tvbot.html)	

Prokaryotic expression analysis of PtCYP721A57 protein

For prokaryotic expression of PtCYP721A57, the primer pair CYP721A57-2-F/R (Table 2) was used for polymerase chain reaction (PCR) amplification, introducing BamHI and HindIII restriction sites upstream and downstream of the gene. After purification and recovery of the target fragment using a DNA gel recovery kit, the fragment was ligated into the pMD19-T vector and transformed into E. coli DH5α competent cells. Positive clones were selected and sequenced by Aoke Biotechnology Co., Ltd., Beijing, China. The PtCYP721A57 gene fragment and the pET28 vector were digested with BamHI and HindIII, respectively. The resulting gene and vector fragments were recovered and ligated overnight at 16 °C. The recombinant plasmid was then transformed into E. coli BL21 (DE3) cells. After confirmation by sequencing, monoclonal cells were selected and cultured at 37 °C with shaking at 200 rpm until reaching an OD600 of 0.6–0.8 (approximately 12 h). Subsequently, the recombinant protein and the corresponding empty vector bacterial solutions were divided into two groups and cultured at different temperatures (16 and 25 °C). Each temperature group included an empty vector control, a non-induced recombinant protein group, and groups induced with varying concentrations of Isopropyl β-D-1-thiogalactopyranoside (IPTG: 0.1, 0.2, 0.5, 1, 2 mM). Following induction, 1–2 mL samples were collected for subsequent sodium dodecyl sulfate-polyacrylamide gel electrophoresis (SDS-PAGE) analysis. The remaining bacterial solution was centrifuged at room temperature to collect cells (5,000 rpm, 6 min). The cells were then resuspended in 10 mL phosphate-buffered saline (PBS) and subjected to two additional rounds of centrifugation and resuspension in PBS. Cell disruption was achieved on ice for 10 min using a high-pressure disruptor. The lysate was then centrifuged at 4 °C (12,000 rpm, 20 min), and 1 mL of the supernatant was collected for SDS-PAGE analysis.

Quantitative real-time polymerase chain reaction (qRT‑PCR) analysis

The expression levels of the PtCYP721A57 gene in different parts of P. tenuifolia and its response to NaCl, PEG6000 stress, and exogenous CHT and ABA treatments were analyzed using qRT-PCR. The primer pair q-PtCYP721A57-F/R (Table 1) was used for PCR amplification. Relative expression of PtCYP721A57 in each plant part and under various treatment time points was determined based on the Ct value at the plateau stage of quantitative amplification curves. Glyceraldehyde-3-phosphate dehydrogenase (GAPDH) was a normalization reference gene (Gao et al., 2012). The StepOnePlus™ Real-Time PCR instrument (Applied Biosystems, Waltham, MA, USA) was employed for these analyses. The relative expression level of the PtCYP721A57 gene was calculated using the 2−ΔΔCt method (Li et al., 2019).

Statistical analysis

Image processing and t-test analysis were conducted using Prism 9.0 (GraphPad Software, La Jolla, CA, USA) and SPSS 27.0 (IBM Corp., Armonk, NY, USA) software packages. A significance level of P < 0.05 was used to determine statistical significance.

Results

Cloning of the PtCYP721A57 gene

Following the sequencing and annotation of the full-length transcriptome of P. tenuifolia, a PtCYP450 gene was identified. Subsequent PCR amplification, sequencing analysis, and NCBI alignment confirmed the presence of the P. tenuifolia gene CYP721A57. This gene possessed an open reading frame (ORF) of 1,521 base pairs, encoding a 506-amino acid protein and contained a conserved CYP450 domain, thereby confirming its classification within the CYP450 family (Figs. 1–3).

Figure 1 The PCR-amplified product of PtCYP721A57.

M, Gene Ruler DNA Ladder Mix; A, PCR amplified product of PtCYP721A57.

Figure 2 Complete cDNA and deduced amino acid sequences of PtCYP721A57 gene.

Figure 3 Domain analysis of PtCYP721A57 protein.

Structural analysis of PtCYP721A57 protein

The protein encoded by PtCYP721A57 underwent a comprehensive analysis, which included determination of its relative molecular mass, isoelectric point (pI), hydrophilicity, prediction of transmembrane domains, and identification of phosphorylation sites.

The protein encoded by PtCYP721A57 had a molecular formula of C2665H4149N713O728S22, totaling 8,277 atoms, and a relative molecular mass of 58.54 kDa. It consisted of a polypeptide chain comprising 506 amino acids. The theoretical isoelectric point (pI) of the protein was calculated to be 9.30, indicating it was alkaline. Additionally, the protein exhibited a lipid index of 64.97 and an instability index (II) of 53.06. Analysis of amino acid hydrophobicity (Fig. S1) revealed an average hydrophilicity coefficient of −0.312 and an instability coefficient of 32.40, suggesting stability as a hydrophobic protein. Understanding the phosphorylation sites and levels of the target protein can provide an important reference for characterizing its properties (Cong, 2020). Prediction of phosphorylation sites in the PtCYP721A57 protein indicated multiple sites, with 23 serine, 31 threonine, and 15 tyrosine (Fig. S2).

The secondary structure of proteins, characterized by the arrangement of the polypeptide backbone atoms, encompassed α-helices, β-sheets, β-turns, and random coils. Predicting this structure provides insights into the relationship between protein structure and its function (Li et al., 2022). For the PtCYP721A57 protein, the predicted secondary structure includes 49.01% α-helix, 12.25% β-sheet, 5.34% extended chain, and 33.40% irregular coil structures, with α-helices being the predominant element (Fig. 4A). Additionally, the three-dimensional structure of PtCYP721A57 was analyzed using SWISS-MODEL online software, employing cytochrome P4504B1 (SMTL ID: 6c94.1) as a template. The analysis indicated 100% coverage of the protein structure (Figs. 4B and 4C).

Figure 4 Structure prediction of PtCYP721A57.

(A) Secondary structure model of PtCYP721A57; (B) prediction of the tertiary structure of PtCYP721A57 protein; (C) tertiary model of PtCYP721A57 protein C: α-helix (purple); extension chain (yellow); β-turn (cyan); random coil (blue); heme domain (red).

Prokaryotic expression analysis

The PtCYP721A57 gene and the pET28a vector underwent double digestion, followed by ligation using T4 ligase. After PCR verification, the target gene was sequenced, and analysis confirmed that the sequence matched the original, indicating the successful construction of the prokaryotic expression vector pET28a-PtCYP721A57.

The pET28a-PtCYP721A57 recombinant vector was introduced into BL21(DE3) cells, with strains transformed using the empty pET28a vector serving as controls. Different concentrations of IPTG were used to induce protein expression. SDS-PAGE analysis following cell disruption (Fig. 5) revealed that compared to both the control groups (blank and non-induced), IPTG significantly stimulated the expression of the recombinant protein. Differences in IPTG concentrations did not significantly affect protein expression, confirming the successful expression of the recombinant plasmid. However, no protein expression was observed in the supernatant, suggesting the absence of soluble protein. The addition of 8M urea facilitated overnight dissolution of the precipitate at 4 °C, leading to detectable expression, and confirming the formation of inclusion bodies. Additionally, it was found that expression in inclusion bodies was higher at 25 °C compared to 16 °C.

Figure 5 Detection of PtCYP721A57 recombinant protein expressed by different temperatures and IPTG concentration.

(A–D) M, Marker; A, pET28a-BL21(DE3); B, pET28a-PtCYP721A57-BL21(DE3); C1–C5, different concentrations of IPTG induced recombinant vector bacterial solution; D1–D5, Supernatant; E1–E5, urea dissolution precipitation.

Subcellular localization analysis

To explore the cellular role of the PtCYP721A57 protein, the recombinant plasmid pBWA(V)HS-PtCYP721A57-GLosgfp was constructed. The localization of the protein within cells was assessed using tobacco transient transformation. Sequencing confirmed the successful ligation of the target gene with the vector, validating the construction of the recombinant subcellular localization vector. Observation of tobacco leaf epidermal cells under a laser confocal microscope revealed green fluorescence signals distributed in the endoplasmic reticulum from the pBWA(V)HS-PtCYP721A57-GLosgfp recombinant vector (Fig. 6).

Figure 6 Subcellular localization of PtCYP721A57 protein in lower epidermal cells of tobacco leaves.

Phylogenetic tree analysis of protein sequences

In this study, the amino acid sequences of CYP450 family members from various plants, including Arabidopsis thaliana (a model plant), and those sharing homology with PtCYP721A57 from Teleostei were retrieved from the NCBI database. Using MEGA 11.0 software, a phylogenetic tree was constructed to analyze the relationship among PtCYP721A57 and other CYP450 family members (Fig. 7). The phylogenetic analysis revealed that proteins encoded by CYP72 subfamily genes formed a distinct branch. This branch showed high homology with sequences from P. tenuifolia, Juglans regia, and Quercus. This clustering suggests potential similar biological functions among the proteins within this branch.

Figure 7 Phylogenetic analysis of PtCYP721A57.

Gene expression analysis

Total RNA was extracted from the roots, xylem, and cortex of P. tenuifolia, and qRT-PCR was used to evaluate the expression of the PtCYP721A57 gene in each tissue. The GADPH gene (housekeeping gene) served as a reference for normalizing PtCYP721A57 expression levels (Fig. 8). Using the equation y = 341,572x − 156,068, the tenuifolin content was calculated for each sample. The cortex exhibited the highest tenuifolin content, followed by the root, while the xylem showed the lowest content (9.594 ± 0.122 mg/g). When using the xylem sample as a reference, the relative expression of the PtCYP721A57 gene correlated with the change in tenuifolin content. Correlation analysis revealed a significant positive correlation between the relative expression of PtCYP721A57 and tenuifolin content, with a correlation coefficient of 0.900 (P < 0.01).

Figure 8 The correlation analysis between the relative expression of the PtCYP721A57 gene in different tissues and the content of tenuifolin was analyzed.

Different capital letters above the column represent the content of tenuifolin in other tissues, and different lowercase letters represent the relative expression levels in different tissues with significant statistical differences (P < 0.05).

Figure 9A illustrates the expression dynamics of the PtCYP721A57 gene at different time points in P. tenuifolia seedlings under various stress and induction conditions, with 0 h (CK) representing the control. Under Mock treatment, PtCYP721A57 expression initially decreased within 6 h, followed by an increase that peaked at 48 h (1.18-fold of CK). In the case of CHT induction, PtCYP721A57 expression showed an initial decline, followed by an increase starting from 12 h, reaching its peak at 48 h (2.84-fold of CK), which was 2.50 and 1.63 times higher than that observed under Mock and CTS treatments at the same time point, respectively. Similarly, under ABA treatment, PtCYP721A57 expression initially decreased and then increased, peaking at 48 h (1.75-fold of CK), which was 1.54 and 0.61 times higher than that observed under Mock and CHT treatments at the same time point, respectively.

Figure 9 Expression pattern of PtCYP721A57 in response to hormones (A) and stress treatments (B).

CHT, Chitosan; ABA, Abscisic acid; Mock groups in A and B are the corresponding hormone and stress treatment mock groups, respectively. Different lowercase letters (a, b, c) above the columns represent statistically significant differences among different hormones and different stresses; different capital letters represent statistically significant differences between different processing times (P < 0.05).

Furthermore, Fig. 9B depicts the expression profile of PtCYP721A57 under different stress treatments at various time points. Under PEG6000 treatment, the relative expression of PtCYP721A57 gradually increased over the first 24 h, reaching its peak at 48 h (1.29-fold of CK), which was 1.05 and 1.89 times higher than that observed under Mock and NaCl treatments at the same time point, respectively. In contrast, NaCl treatment for 6 h significantly induced PtCYP721A57 expression (3.16-fold of CK), which was 11.42 and 3.17 times higher than that observed under Mock and PEG treatments simultaneously.

Discussion

The CYP450 family genes play crucial roles in the metabolic pathways of both endogenous and exogenous substances across a wide range of organisms. Currently, 127 CYP450 families have been found in plants, which are classified into 11 clusters showing distinct evolutionary branches on phylogenetic trees (Ghosh, 2017). These genes are known to catalyze various biological processes including growth, development, and the synthesis of terpenoids and sterols. In recent years, there has been increasing interest in the screening and exploration of P450 family genes, particularly their functional roles in the biosynthesis pathways of triterpenoid saponins in medicinal plants (Das et al., 2023).

In this study, the complete cDNA sequence of the PtCYP721A57 gene was cloned via PCR, followed by bioinformatics analyses. The results confirmed that PtCYP721A57 possesses a full ORF and harbors a conserved domain characteristic of the P450 superfamily, thereby classifying it as a member of this family. The PtCYP721A57 protein was identified as hydrophobic and stable. Subcellular localization assays indicated predominant localization of PtCYP721A57 within the endoplasmic reticulum. To facilitate further study, a prokaryotic expression vector, pET28a-PtCYP721A57, was constructed and introduced into BL21(DE3) cells for induction. The results demonstrated successful expression of the PtCYP721A57 recombinant plasmid at different temperatures and IPTG concentrations. Following sonication, gene expression was achieved in BL21(DE3), with significantly increased protein expression after IPTG induction. Phylogenetic tree analysis, mainly used to study the classification and evolution of gene family members, provided insights into the biological functions of closely related protein family members (Zhao et al., 2023). Molecular phylogenetic analysis of PtCYP721A57 and 10 other species within the CYP72 subfamily revealed its affiliation with the CYP72A subfamily. PtCYP721A57 clustered primarily with GrCYP734A1, QlCYP734A1, and QsCYP734A1 from legumes, suggesting potential shared biological functions among these proteins. According to Biazzi et al. (2015), the synthesis of oleanane triterpenoid saponins typically involves CYP72 and/or CYP85 family clusters, implying the potential involvement of PtCYP721A57 from P. tenuifolia in the biosynthesis of oleanane triterpenoid saponins.

P450 genes are pivotal in the regulation of various metabolic reactions in plants, and crucial for normal growth, development, and responses to environmental stresses. Regulation of P450 gene expression primarily occurs at the transcriptional level and exhibits tissue-specific specificity. Given the roots’ interaction with soil microecology, distinct expression responses of P450 genes are anticipated, potentially aiding plant adaptation and the synthesis of secondary metabolites (Gao et al., 2023). At present, the medicinal use of P. tenuifolia categorizes products into three specifications based on initial processing, with cortex-containing specimens commanding higher prices (Zhang et al., 2017; Wang, Peng & Hu, 2017). Analysis in this study detected higher tenuifolin levels in the cortex, correlating with prominent PtCYP721A57 expression in this tissue. This suggests potential localization of tenuifolin within the phloem of P. tenuifolia. Endogenous hormones play important roles in signal transduction, regulating plant physiological and biochemical reactions, with CYP450 gene expression closely related to these processes. PtCYP721A57 responded significantly to ABA and CHT inductions, showing increased expression levels after hormone treatment. Additionally, PtCYP721A57 responded to drought and NaCl stresses, indicating potential roles in mediating physiological and metabolic processes in P. tenuifolia. Through gene cloning and bioinformatics analyses, preliminary insights indicate that the CYP721A57 gene may participate in downstream reactions of polygala oleanane triterpenoid saponin synthesis. However, the specific regulatory pathways and mechanisms need to be further studied. These findings enhanced our understanding of the functions of CYP450 genes involved in the synthesis of oleanane triterpenoid saponins in P. tenuifolia, providing scientific foundations for their regulation and genetic improvement.

Conclusion

In this study, the PtCYP721A57 gene from P. tenuifolia was cloned via PCR, and comprehensive analyses including bioinformatics, the accumulation of tenuifolin content across different tissues, response to hormones and abiotic stress, subcellular localization, and prokaryotic protein expression were conducted. These findings contributed to a deeper understanding of the pivotal role of the CYP450 gene in the defense responses of P. tenuifolia. They also provided evidence for elucidating the biosynthetic pathways and regulatory mechanisms of P. tenuifolia saponins. Importantly, this study laid the groundwork for enhancing the quality of medicinal materials through genetic engineering and offered a basis for further investigations into the functions of CYP450 genes in P. tenuifolia.

Supplemental Information

Supplemental Information 1 Raw qRT-PCR data.

Supplemental Information 2 Hydrophobicity prediction analysis of amino acid sequence of PtCYP721A57.

Supplemental Information 3 Prediction of phosphorylation sites of PtCYP721A57.

Additional Information and Declarations

Competing Interests

Author Contributions

Data Availability

The authors declare that they have no competing interests.

Yao Luo conceived and designed the experiments, prepared figures and/or tables, and approved the final draft.

Benxiang Hu conceived and designed the experiments, authored or reviewed drafts of the article, and approved the final draft.

Haiyue Ji performed the experiments, analyzed the data, prepared figures and/or tables, and approved the final draft.

Yiyao Jing performed the experiments, analyzed the data, prepared figures and/or tables, and approved the final draft.

Gang Zhang conceived and designed the experiments, analyzed the data, prepared figures and/or tables, and approved the final draft.

Yonggang Yan conceived and designed the experiments, prepared figures and/or tables, and approved the final draft.

Bingyue Yang analyzed the data, authored or reviewed drafts of the article, and approved the final draft.

Liang Peng conceived and designed the experiments, authored or reviewed drafts of the article, and approved the final draft.

The following information was supplied regarding data availability:

The qRT-PCR data is available in the Supplemental File.

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
