# Peer review of "Sequence characteristics, expression and subcellular localization of PtCYP721A57 gene from cytochrome P450 family in Polygala tenuifolia willd"

_PeerJ, doi:10.7717/peerj.18089_

## Round 0.1 · original submission · Major Revisions

Please address concerns of all reviewers and revise manuscript accordingly.

Reviewer 1 ·

Basic reporting

This study focused on the PtCYP721A57 gene from Polygala tenuifolia, revealing that its 1521 bp ORF encodes a 506-amino acid hydrophobic protein localized in the endoplasmic reticulum. Expression analysis indicated that PtCYP721A57 is highly expressed in the cortex, root, and xylem, with varying responses to hormonal and abiotic stress, notably showing increased expression under ABA, chitosan, drought, and NaCl treatments. These findings provide a foundation for exploring PtCYP721A57's role in the synthesis of triterpenoid saponins in P. tenuifolia.
1. The abstract contains too many details. The authors should describe it more concisely and briefly.
2. Line 207 appears to be a figure legend. The authors should move it to the appropriate figure legend section.
3. The descriptions in lines 240-243 should be moved to the figure legends and methods section, not the results section.
4. The authors should use high-resolution images. Please fix Figures 3, 9, and 10. Additionally, Figure 10 was mistakenly labeled as Figure 9 in the manuscript.
5. The gel images in Figure 5 look unclear. The authors should adjust the contrast or improve the gel quality in the experiments.
6. All figures need more detailed legends, not just a simplified sentence.
7. Figure 8 needs a description for the y-axis.

Experimental design

8. Line 157 contains unrecognized symbols. The authors should correct these.
9. The authors have not specified the statistical methods used to determine significance. They should clearly state these methods.

Validity of the findings

no comment

Reviewer 2 ·

Basic reporting

I think the author did not carefully prepare this manuscript, and many pictures are almost indistinct, such as fig 9. Why is there several pictures in a picture, such as fig 5, without annotations or illustrations, I don't know what the author is saying. I suggest the author to prapare the ms according to the guidelines strictly to review. fig8 is confusing. shorten line 95, 115 and 143, it is redundancy.

Experimental design

It is simple.

Validity of the findings

I did not find the novelty.

Reviewer 3 ·

Basic reporting

See below

Experimental design

See below

Validity of the findings

See below

Additional comments

The authors reported the cloning, identification, and analysis of the PtCYP721A57 gene in Polygala tenuifolia, laying the groundwork for further investigations into its role in the synthesis pathway of triterpenoid saponins in P. tenuifolia. I think this study provides significant insights for the reader. My comments may help improve this manuscript and enhance its quality for publication. I recommend a major revision.

The pharmacological effects of P. tenuifolia should be clarified in more detail in the introduction.

The quality of all figures needs to be improved with higher resolution.

The discussion should be more in-depth to provide significant contributions to this study.

Many errors were found in the reference section. Please correct them to match the style of this journal.

Finally, English in this manuscript need to be carefully checked by native speaker.

---

## Round 0.2 · accepted · Accept

In my view, all the issues pointed by the reviewers were adequately addressed and the manuscript was revised accordingly. I respectfully disagree with comments of the reviewer #2 and think that the manuscript was revised in line with the original queries of the reviewers. Therefore, I am accepting revised manuscript in its present form.

Reviewer 1 ·

Basic reporting

The authors have addressed all the questions.

Experimental design

The authors have addressed all the questions.

Validity of the findings

The authors have addressed all the questions.

Reviewer 2 ·

Basic reporting

no comment

Experimental design

no comment

Validity of the findings

no comment

Additional comments

Please take the external review comments seriously. The author did not revise according to the comments, such as the line 198- line202, it is the method not the result. Therefore, I suggest the authors to revised the ms from a professional perspective. Otherwise, the ms should be reject, it is simple and not meet the journal requirement.